# Testing AIware Systems: A Software Engineering Survey

### Karla Gonzalez
Royal Military College of Canada
Kingston, Canada
karla.gonzalez@rmc-cmr.ca

### Mariam El Mezouar
Royal Military College of Canada
Kingston, Canada
mariam.el-mezouar@rmc.ca

## Abstract

Foundation models, particularly large language models, are increasingly embedded as core components of software systems. This shift has given rise to a growing body of research on testing such systems, referred to in this paper as AIware systems. While prior work proposes numerous techniques to expose undesirable behaviors, it remains unclear how these approaches align with established software testing practices and support the software lifecycle.

This survey analyzes the AIware testing literature through the lens of classical software engineering concepts. We examine testing levels, oracle strategies, automation readiness, and diagnostic support, and assess how existing approaches map to lifecycle activities such as integration testing, regression testing, and CI-integrated workflows. Our results show that the literature is strongly concentrated on system-level, pre-release evaluation, with limited operational support for integration, regression, and deployment-time testing.

We further show that many of these gaps stem from fundamental challenges in oracle design, including non-determinism, underspecified correctness, and limited diagnosability. Without stable and automatable decision criteria, AIware testing techniques remain difficult to integrate into continuous development and maintenance pipelines.

Overall, this survey provides a structured characterization of the current state of AIware testing research and identifies key structural challenges that must be addressed to support lifecycle-aware, reliable AIware systems.

## CCS Concepts

• **Software and its engineering** → **Software verification and validation**; • **Computing methodologies** → *Natural language processing*; Machine learning.

## Keywords

AIware systems, Foundation models, Software testing, Test oracles

**ACM Reference Format:**
Karla Gonzalez and Mariam El Mezouar. 2026. Testing AIware Systems: A Software Engineering Survey. In *Proceedings of the 3rd ACM International Conference on AI-Powered Software (AIware '26), July 6–7, 2026, Montreal, QC, Canada.* ACM, New York, NY, USA, 16 pages. https://doi.org/10.1145/3805760.3814894

## 1 Introduction

Foundation models (FMs), particularly large language models (LLMs), are increasingly embedded as core components of software systems [1, 7, 8]. These systems—referred to here as *AIware systems*—combine prompt templates, orchestration logic, retrieval components, external tools, and agentic workflows. Their behavior emerges from interactions between conventional software modules and probabilistic model components, rather than deterministic code alone.

This shift challenges core assumptions of classical software testing. Traditional testing practices rely on determinism, stable interfaces, and well-defined test oracles [6, 25]. Given the same input, a system is expected to produce the same output, enabling unit testing, integration testing, regression testing, and automated release gates [13, 28, 43]. In AIware systems, however, non-deterministic outputs [3], prompt sensitivity [27], and evolving external dependencies complicate the direct application of these practices.

In response, a growing body of work proposes testing techniques targeting AIware systems, including robustness and perturbation-based testing [38–41], search-based test generation [30, 44], metamorphic testing [9, 15], and human-in-the-loop evaluation [20]. However, this emerging literature is uneven. Most approaches emphasize end-to-end system behavior and pre-release evaluation, while integration testing, regression testing, CI-integrated workflows, and debugging support receive limited operational treatment [16, 46]. Testing is frequently framed as evaluation or benchmarking rather than as a lifecycle activity embedded in software development.

As a result, it remains unclear how AIware testing research aligns with established software engineering practice. Specifically, we lack a structured view of: (1) how existing work maps to classical testing levels, (2) which foundational testing assumptions hold or break down in AIware settings, and (3) whether current approaches support repeatable, automation-ready testing across the software lifecycle.

Prior surveys have examined testing of machine learning models at the model level [46] and behavioral testing for NLP systems [27]. However, neither examines how testing research maps to software engineering lifecycle practices—such as integration testing, regression testing, and CI-based quality control—when FMs are embedded as components within larger software systems. To address this gap, we adopt a testing-first perspective rooted in software engineering. Rather than organizing prior work by application domain or model architecture, we analyze the literature through the lens of classical testing concepts and lifecycle stages. Our goal is to characterize where testing activity is concentrated, identify structural gaps, and examine how AIware properties reshape established testing assumptions.

This survey is guided by the following research questions:

**RQ1:** How is existing research on testing AIware systems structured in terms of testing levels, system types, and testing techniques?
**RQ2:** Which assumptions underlying classical software testing hold, weaken, or break down in AIware systems?
**RQ3:** To what extent do existing AIware testing approaches provide support for core software testing activities across the software lifecycle?

By answering these questions, we make three contributions. First, we provide a systematic mapping of AIware testing research to classical testing levels. Second, we analyze how key testing assumptions—such as determinism, oracle reliability, and failure attribution—are affected in AIware systems. Third, we synthesize lifecycle implications, identifying limitations in integration, regression, and CI-integrated testing and highlighting oracle design as a central bottleneck for practical adoption.

As AIware systems become integral to production environments, establishing testing practices that align with software engineering principles is essential for supporting long-term reliability and system evolution.

## 2 Survey Methodology & Corpus Construction

This survey follows a structured literature review approach. The goal is not to exhaustively review all work related to foundation models, but to analyze existing *software testing approaches* for AIware. We focus on identifying which testing activities are currently supported, which assumptions no longer hold, and which parts of the software testing lifecycle remain underexplored within the emerging AIware testing landscape.

The goal of this search process is to identify studies that contain concrete evidence of testing activity for AIware systems. Papers are treated as individual units of analysis. The behaviours we search for are mainly described through methodologies, evaluations and reported results, rather than conceptual discussions. Subsequent screening and coding decisions focus on whether a paper has sufficient evidence and detail to support explicit classification along our dimensions.

### 2.1 Scope of the Review

We study *AIware systems*, defined as software systems that integrate foundation models (e.g., large language models) as core components alongside conventional software modules such as orchestration logic, retrieval mechanisms, external tools, and user interfaces.

The scope of this survey covers testing and evaluation of AIware systems from a software engineering perspective. This includes studies that propose or analyze concrete testing activities at any level—unit, integration, system, regression, or deployment—provided they target software systems that embed FMs, as opposed to model-only evaluation or benchmark-only assessment. We also examine whether existing work provides support for debugging, regression testing, or lifecycle-integrated testing, without assuming that such support exists.

A system qualifies as AIware under this definition when a foundation model is embedded as a core component whose probabilistic and generative properties influence system behavior. This distinguishes AIware from traditional systems that may use an LLM as a drop-in replacement behind a fixed, deterministic adapter (e.g.,

**Table 1: Study inclusion and exclusion criteria**

| Inclusion criteria (all must hold): |
| --- |
| ✓ The study examines AIware systems or provides a structured analytical framework (e.g., taxonomy) grounded in such systems. |
| ✓ The study addresses testing or evaluation from a system-level software engineering perspective. |
| ✓ The study proposes, analyzes, or systematically categorizes testing activities, test oracles, or testing workflows. |
| ✓ The study goes beyond isolated model training or benchmark-only evaluation. |
| ✓ The study provides concrete methods rather than high-level visions, roadmaps, or challenge catalogues. |

| Exclusion criteria (any one is sufficient): |
| --- |
| ✗ The study focuses exclusively on foundation model architecture, training, or pretraining. |
| ✗ The study reports benchmark results without discussing system-level testing implications. |
| ✗ The study uses foundation models only to support traditional software engineering tasks (e.g., test generation). |
| ✗ The study discusses reliability or trustworthiness only at a conceptual level without operational testing methods. |

returning only a top-1 classification label). In the latter case, assumptions such as deterministic execution (A1) and oracle reliability (A3) may be less severely affected, because the model's stochastic behavior is not exposed to the rest of the system. By contrast, systems where the FM participates in open-ended generation, multi-step reasoning, or tool invocation expose more of the properties that challenge classical testing assumptions. The degree of assumption breakdown thus depends on how much of the FM's probabilistic nature propagates through the system architecture.

Table 1 summarizes the inclusion and exclusion criteria used in this survey. Papers that discussed testing challenges or risks without proposing structured frameworks (e.g., taxonomies) or concrete testing procedures were excluded. The resulting corpus therefore consists of studies that provide sufficient methodological and evaluative detail to support evidence-based classification of testing characteristics, which forms the basis of analyses performed under each research question.

### 2.2 Search Strategy and Study Selection

The literature search and corpus construction follow an iterative, multi-phase procedure summarized in Table 2. The procedure is designed to identify software testing methodologies for AIware systems while maintaining a controlled and reproducible search scope. Coding was performed by one researcher. A second researcher independently reviewed all coding and screening decisions, and disagreements were resolved through collaborative discussion.

*Search sources and scope.* The search is conducted using Google Scholar, the ACM Digital Library, IEEE Xplore, and arXiv. Google Scholar serves as the primary discovery engine due to its broad coverage of venues and preprints, while the remaining sources are

**Table 2: Iterative literature search, screening, and gap analysis procedure**

| Phase | Step | Action | Outcome |
|---|---|---|---|
| Seed discovery | Query definition | Define initial keyword queries based on the research scope. | Query set |
| | Keyword search | Run keyword-based searches using academic search engines. | Initial candidate papers |
| | Initial screening | Screen titles and abstracts using inclusion and exclusion criteria. | Seed paper set |
| Citation expansion | Backward search | Examine references of each seed paper. | Additional candidates |
| | Forward search | Examine papers citing each seed paper. | Additional candidates |
| | Related search | Review closely related papers suggested by the search engine. | Additional candidates |
| | Screening | Screen expanded candidates using the same criteria. | Expanded corpus |
| Gap analysis | Coverage assessment | Assess coverage across testing levels and lifecycle stages. | Identified gaps |
| | Targeted search | Run focused searches targeting the identified gaps. | Gap-specific candidates |
| | Re-screening | Screen gap-specific candidates using the same criteria. | Updated corpus |
| Stopping criteria | Saturation check | Assess whether new papers introduce new testing concepts. | Corpus saturation |
| Finalization | Corpus freeze | Finalize the paper set used for synthesis. | Final corpus: **AIware testing corpus** |

used to confirm coverage in core software engineering and systems venues.

For all keyword-based and targeted searches, results were screened until thematic saturation was observed. In practice, this typically occurred within the top 10 ranked results, after which no additional relevant studies meeting our inclusion criteria were identified. This is consistent with the observation that for focused research topics, the most relevant results cluster within the first page of academic search engines. The top-10 limit applies only to keyword-based discovery (Phases 1 and 3); citation-based expansion in Phase 2 imposes no such limit. For citation-based exploration, the set of papers considered is determined by the source paper itself. In practice, this corresponds to examining reference lists averaging approximately 40 references per seed paper. Forward citation search is applied when available; however, many recent papers have low citation counts, which naturally limits the size of the forward citation set.

*Phase 1: Seed discovery.* In the seed discovery phase, 15 keyword queries were defined based on three query families: (i) testing- and evaluation-oriented terms (e.g., "LLM application testing", "testing LLM-based systems", "foundation model testing", "debugging LLM systems"), (ii) system- and architecture-oriented terms (e.g., "agentic LLM evaluation", "retrieval augmented generation evaluation", "tool-using LLM evaluation"), and (iii) reliability- and deployment-oriented terms (e.g., "LLM reliability production", "behavioral evaluation of LLM systems", "LLM system robustness"). The complete query list with rationale is provided in the replication package.[1]

For each query, the top 10 search results were screened at the title and abstract level using the inclusion and exclusion criteria described in Table 1. Phase 1 screened 150 candidates (10 per query × 15 queries), of which seven papers satisfied the screening criteria and were forwarded as seed papers to the next phase.

*Phase 2: Citation expansion.* In the citation expansion phase, the seven seed papers are used as starting points for backward and

forward citation exploration, as well as similarity-based expansion using related-article suggestions provided by the search engine. Backward citation search examines the reference list of each seed paper, while forward citation search examines papers that cite the seed paper when such information is available.

Candidate papers identified during citation expansion are screened immediately using the same title and abstract criteria applied in Phase 1. This conservative expansion strategy limits topic drift and ensures that only papers closely aligned with the survey scope are retained for further consideration. Phase 2 screened 479 candidates across all seed papers, retaining 14 additional papers. The majority of the final corpus was identified through this phase, which imposed no limit on the number of results considered per seed paper.

*Phase 3: Gap analysis.* After constructing the core corpus, a final search phase verifies coverage across classical software testing levels. The corpus was classified by testing level, and underrepresented levels were identified (specifically: regression testing, CI-integrated testing, and debugging/fault localization). Three targeted queries were run to search for work addressing these gaps: "Regression testing LLM-based software", "continuous integration testing LLM application", and "debugging OR fault localization LLM-based system". As in Phase 1, screening was limited to the top 10 results per query and used the same inclusion and exclusion criteria. Phase 3 screened 30 candidates and retained no new papers, confirming that the identified gaps reflect substantive research absence rather than limitations of the initial search strategy.

*Screening and corpus finalization.* Screening decisions are applied throughout all phases of the procedure to determine whether candidate papers are forwarded or excluded. Across the full process, 662 candidate papers were screened at least at the title or abstract level. 21 of these passed title-and-abstract screening across the three phases (7 from Phase 1, 14 from Phase 2, and 0 from Phase 3); a subsequent full-text review excluded 5 that did not meet the inclusion criteria upon close reading. These 16 studies form the final corpus, which we refer to in the rest of the paper as the *AIware testing corpus.* An additional six foundational papers (e.g., on

---

[1]All replication materials, including search logs, coding sheets, and RQ analysis datasets, are publicly available at: https://figshare.com/s/de28656ebb590a85b2ce

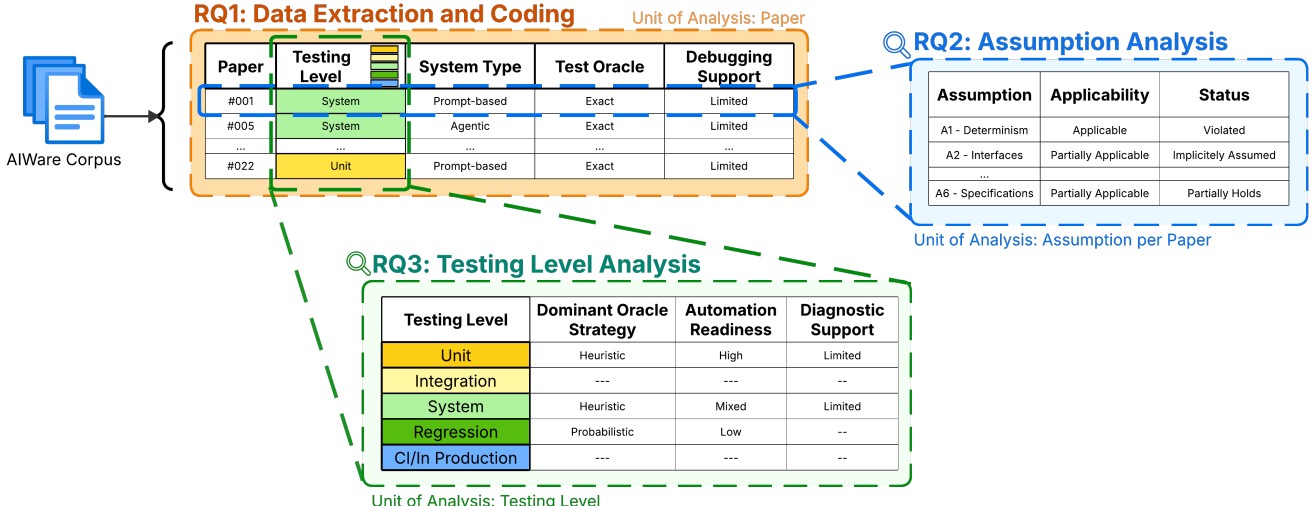

**Figure 1: Overview of the study design and unit of analysis across research questions. RQ1 operates at the paper level, extracting and coding study attributes (testing level, system type, test oracle, and debugging support). RQ2 maps six classical testing assumptions to each paper, with the unit of analysis being an assumption per paper. RQ3 aggregates results by testing level to synthesize dominant oracle strategies, automation readiness, and diagnostic support across the AIware testing corpus.**

metamorphic testing, chaos engineering, and fault injection) were used for conceptual framing but are not part of the primary empirical corpus. The corpus was considered saturated when additional rounds did not introduce studies that met the inclusion criteria or contributed new testing concepts. Figure 1 summarizes how the three research questions build on the collected corpus and differ in their unit of analysis.

## 3 RQ1 Analysis and Results

RQ1 characterizes the retained AIware corpus along a set of dimensions derived from classical software testing concepts. The goal is descriptive: to identify how testing is currently conducted, what levels are targeted, what techniques are employed, and how correctness and failure are evaluated.

Table 3 defines the coding dimensions used in this analysis. In addition to these core testing dimensions, we also recorded descriptive corpus characteristics, including publication year, venue type, and the primary testing technique employed by each study. These attributes are used to summarize the maturity and methodological distribution of the field but are not part of the formal testing-dimension taxonomy defined in Table 3. Detailed per-paper coding examples are provided in Appendix A (Tables A1 and A2).

## 3.1 Coding and Data Extraction

For each retained study, one row was recorded in a structured extraction spreadsheet available in the replication package introduced in Section 2.2. Coding decisions were based on explicit evidence in the methodology, evaluation, and results sections of each paper. Testing level was assigned by identifying the artifact under test and the execution scope. AIware system type was determined from the architectural role of the foundation model during evaluation. Test

oracle type was derived from the mechanism used to determine acceptable behavior. Debugging support was coded based on whether the study provided diagnostic insight beyond failure reporting. The primary testing technique was identified from the study's main methodological contribution (e.g., robustness testing, search-based generation, taxonomy-driven analysis, stress testing, or coverage criteria).

Each paper was assigned one primary value per dimension, to reflect its main contribution. Secondary aspects were noted internally but not counted toward aggregate summaries. Appendix A (Tables A1 and A2) provides detailed coding examples for representative studies to illustrate how classification decisions were linked to textual evidence. The findings reported in this paper are limited to the retained corpus (N=16) and do not extend beyond this evidence-based set of studies.

## 3.2 Corpus Overview

Table 4 lists all 16 studies retained in the AIware testing corpus, along with their publication year, venue type, primary testing level, and system type. This table serves as a reference for the analyses that follow.

## 3.3 Results

**Publication year.** The corpus is dominated by very recent work, with all 16 retained studies published between 2024 and 2026. This suggests that systematic testing of AIware systems is an emerging research area rather than a mature subfield.

**Venue type and peer-review status.** The literature spans conferences, journals, and preprint servers. Of the 16 retained studies,

**Table 3: Coding dimensions used for data extraction and analysis**

| Dimension | Value | Definition |
|---|---|---|
| Testing level | Unit | Testing individual components in isolation, such as prompt templates, scoring functions, or wrapper code around a foundation model. |
| | Integration | Testing interactions between multiple components, such as a model combined with retrieval, tools, or orchestration logic. |
| | System | End-to-end testing of the complete AIware system, focusing on overall behavior from inputs to outputs. |
| | Regression | Testing whether system behavior changes unintentionally after updates to prompts, models, data, or system configuration. |
| | CI | Testing activities designed to run automatically as part of a continuous integration or deployment pipeline. |
| | In-production | Testing or validation performed on deployed systems using live or shadow traffic. |
| System type | Prompt-based | Systems where behavior is primarily driven by prompt design and prompt templates. |
| | RAG | Retrieval-augmented generation systems that combine a foundation model with external knowledge sources. |
| | Agentic | Systems that perform multi-step reasoning or decision-making using planning, memory, or action loops. |
| | Tool-using | Systems where the foundation model interacts with external tools or APIs to complete tasks. |
| Test oracle | Exact | Oracles with a clearly defined correct output or assertion, enabling deterministic pass/fail decisions. |
| | Heuristic | Oracles based on expectations, thresholds, or qualitative criteria rather than exact correctness. |
| | Metamorphic | Oracles that check consistency across related inputs or executions instead of absolute correctness. |
| | Human | Oracles that rely on human judgment, ratings, or rubric-based evaluation. |
| Debugging support | None | The approach reports failures without providing diagnostic information about their cause. |
| | Limited | The approach provides aggregate signals such as failure counts, coverage metrics, or performance trends. |
| | Strong | The approach supports fault localization, causal analysis, or explicit diagnosis of failure sources. |

12 appear as preprints and 4 have undergone peer review in established software engineering venues, including AST, ICSTW, CAIN, and the *Journal of Systems and Software* [23, 35, 42, 44]. The predominance of preprints reflects the rapid evolution of AIware testing research. The same inclusion criteria were applied uniformly across venues.

**Testing levels.** Within the retained corpus, 13 of 16 studies conduct system-level evaluation as their primary testing activity. These studies evaluate end-to-end AIware behavior, exercising complete workflows, prompt–example combinations, or multi-component interactions [3, 11, 15, 18, 20, 30, 33, 35, 38–41, 44]. Regression testing appears in one study, which analyzes prompt behavior under evolving LLM APIs and examines the impact of silent model updates [23]. Unit-level testing is addressed by one study, which proposes model-level adequacy and coverage criteria that operate at the level of internal LLM components rather than full application workflows [42]. One study [32] contributes a defect taxonomy without targeting a specific testing level. Some infrastructure-oriented studies also analyze bugs and testing gaps at the component or integration level [18, 35], but these are not framed as isolated unit-testing methodologies.

We did not identify any retained study that treats CI-integrated or in-production testing as a primary testing level. Agent reliability studies evaluate production-like stress conditions [15], but they

do so as controlled experiments rather than as part of ongoing development or deployment workflows. This approach mirrors fault injection and chaos engineering, where controlled failures are introduced to evaluate system resilience [2, 5]. Overall, the retained corpus is strongly concentrated on system-level evaluation, with limited representation at the unit and regression levels and no coverage of integration, CI-integrated, or in-production testing.

**AIware system type.** The retained studies cover a range of AIware system types. Prompt-based and LLM-driven application systems account for 10 of 16 studies, particularly in robustness and search-based testing frameworks [3, 30, 32, 38–42, 44]. Human-in-the-loop quality evaluation of prompt-driven applications is also represented [20]. Agentic and tool-using systems account for 6 studies, with emphasis on reliability, stress conditions, tool/API interaction behavior, and infrastructure-level defects [11, 15, 18, 23, 33, 35]. Finally, model-centric testing approaches appear in work proposing coverage and adequacy criteria at the level of internal LLM components rather than complete application workflows [42]. Within the retained corpus, testing challenges arise across multiple layers of the AIware stack, from prompt templates and application logic to agent coordination and infrastructure dependencies.

**Testing techniques employed.** The retained studies can be grouped according to their primary methodological approach to test generation or failure analysis.

**Table 4: Studies retained in the AIware testing corpus (N=16)**

| Study | Year | Venue | Venue Type | Testing Level | System Type |
|---|---|---|---|---|---|
| Yoon et al. [44] | 2025 | ICSTW | Conference | System | Prompt-based |
| Xiao et al. [40] | 2025 | arXiv | Preprint | System | Prompt-based |
| Kathiresan [20] | 2025 | Authorea | Preprint | System | Prompt-based |
| Vinay [33] | 2025 | arXiv | Preprint | System | Agentic |
| Gupta [15] | 2026 | arXiv | Preprint | System | Agentic |
| Xiao et al. [39] | 2024 | arXiv | Preprint | System | Prompt-based |
| Xiao et al. [38] | 2024 | arXiv | Preprint | System | Prompt-based |
| Ma et al. [23] | 2024 | CAIN | Conference | Regression | Tool-using |
| Sorokin et al. [30] | 2026 | arXiv | Preprint | System | Prompt-based |
| Xiao et al. [41] | 2025 | arXiv | Preprint | System | Prompt-based |
| Dobslaw et al. [11] | 2025 | arXiv | Preprint | System | Agentic |
| Atil et al. [3] | 2024 | arXiv | Preprint | System | Prompt-based |
| Jiang et al. [18] | 2025 | arXiv | Preprint | System | Tool-using |
| Tian et al. [32] | 2025 | arXiv | Preprint | — | Prompt-based |
| Winston & Just [35] | 2025 | AST | Conference | System | Tool-using |
| Xie et al. [42] | 2025 | JSS | Journal | Unit | Prompt-based |

- **Robustness and perturbation-based testing.** Four studies focus on generating input variations to expose unstable behavior in AIware systems [38–41]. These approaches apply controlled perturbations or fuzzing strategies to prompts and natural-language inputs and evaluate behavioral changes under variation.

- **Search-based test generation.** Two studies use guided search strategies, such as adaptive random testing or evolutionary optimization, to systematically explore diverse or failure-inducing inputs [30, 44]. These approaches aim to increase failure discovery compared to random input sampling.

- **Taxonomy and failure-model–driven analysis.** Five studies develop structured taxonomies of failures or defects in AIware systems [11, 18, 32, 33, 35]. These studies emphasize systematic categorization of failure modes, oracle ambiguity, and infrastructure-level defects rather than executable test-case generation.

- **Stress and production-oriented evaluation.** Two studies evaluate AIware systems under production-like conditions, including repeated execution, API drift, or injected perturbations [15, 23]. These approaches focus on temporal stability, reliability under faults, and regression across system updates.

- **Coverage and adequacy-based criteria.** One study proposes formalized multi-level coverage criteria to assess testing adequacy at the level of internal LLM components [42]. Unlike robustness testing, this work emphasizes measuring coverage of model behaviors rather than generating adversarial inputs.

- **Non-determinism analysis.** One study empirically investigates output variability under nominally deterministic LLM settings [3], providing foundational evidence for the need for distribution-based test oracles.

**Test oracles.** The studies use a range of oracle strategies to determine acceptable behavior. Of the 16 studies, seven rely on exact or assertion-based oracles, typically using ground-truth labels or deterministic state predicates [3, 15, 18, 33, 35, 40, 44]. Four use heuristic or metric-based oracles, such as error rates, task accuracy, or safety indicators [11, 30, 38, 41]. One study uses metamorphic oracles that assess consistency across input transformations [39]. One study relies on human judgment in a human-in-the-loop evaluation setting [20]. One study uses an LLM-based judge as an automated oracle [42]. Regression-oriented work compares outputs across system versions to detect behavioral drift [23]. To summarize, oracle design in the corpus ranges from exact output matching and heuristic thresholds to metamorphic relations, human evaluation, and model-based judgment.

**Debugging support.** Debugging support is predominantly limited across the corpus. Of the 16 studies, 13 provide only limited diagnostic support, primarily reporting aggregate signals such as failure rates, robustness scores, or coverage metrics without explaining why failures occur [3, 11, 15, 18, 30, 32, 33, 38–42, 44]. Two studies provide strong diagnostic support: human-in-the-loop evaluation collects qualitative feedback to understand failure behavior [20], and Winston & Just map failures to root causes across LLM, tool selection, and parsing components [35]. One study provides limited support through exploratory regression analysis without fault localization [23]. Overall, across the retained corpus, the level of debugging support ranges from aggregate failure reporting to structured root-cause analysis, but strong diagnostic support remains rare.

> **RQ1 Summary**
>
> Testing activity in the AIware corpus (N=16) is strongly concentrated at the system level (13 studies), with unit, regression, integration, CI, and in-production testing rarely or never addressed. Oracle strategies are diverse but debugging support is predominantly limited (13 of 16 studies).

# 4 RQ2 Analysis and Results

The findings of RQ1 indicate that the retained corpus of AIware testing research relies largely on system-level and exploratory techniques. RQ2 examines how foundational assumptions from software engineering testing theory hold when such models are integrated into AIware systems.

## 4.1 Baseline Assumptions from Classical Software Testing.

To analyze how AIware systems challenge traditional testing practice, we make explicit a set of baseline assumptions implicit in classical software engineering testing theory. These assumptions are drawn from established work on test oracles, regression testing, test adequacy, and fault detection, and serve as reference points for analyzing where existing testing approaches break down in AIware systems. They are not intended to be exhaustive, but represent core foundations repeatedly relied upon by classical testing techniques. The analysis in this section is based solely on the retained AIware testing corpus.

**Assumption A1: Deterministic Execution.** Classical software testing assumes that software systems exhibit deterministic behavior: given the same inputs and execution conditions, a system produces the same observable outputs. This assumption is implicit in foundational definitions of testing, where test cases are specified as combinations of inputs, execution conditions, and expected results that are assumed to be reproducible [6, 25].

Deterministic execution is a prerequisite for regression testing. Regression testing techniques rely on re-executing previously passing tests to detect unintended behavioral changes introduced by code modifications, implicitly assuming that test outcomes are stable across executions [28, 43]. The same assumption underlies automated testing and continuous integration practices, where test failures are treated as reliable indicators of software regressions rather than execution variability [13].

> **A1 — Deterministic Execution.** Given the same inputs and system configuration, a software system produces the same observable behavior across repeated executions.

**Assumption A2: Stable and Explicit Interfaces.** Classical software testing assumes that software systems are decomposed into components that expose *stable and explicit interfaces*. Foundational work on modular design emphasizes that systems should be structured around well-defined interfaces that separate external behavior from internal implementation, enabling independent development and reasoning about components [26, 31].

This assumption is central to software testing practice. Testing techniques are designed to exercise components through their public interfaces, treating these interfaces as fixed points of interaction while internal implementations evolve [6]. Test cases are specified in terms of inputs and observable outputs exposed by these interfaces, implicitly assuming that the interface remains stable across executions and versions [25]. More formal approaches, such as design by contract, further treat interfaces as behavioral contracts whose preconditions and postconditions can be validated through testing [24].

> **A2 — Stable and Explicit Interfaces.** Software components expose stable, well-defined interfaces through which their behavior can be exercised and validated independently of their internal implementation.

**Assumption A3: Existence of a Reliable Test Oracle.** Classical software testing assumes the existence of a *reliable test oracle*: a mechanism by which the correctness of a program's behavior can be determined for a given test case. In traditional testing, expected outputs are derived from specifications, requirements, or known correct implementations, allowing test executions to be classified as pass or fail [6, 25].

The importance of test oracles has long been recognized in the testing literature [34]. When exact expected outputs are unavailable or impractical to specify, alternative oracle strategies—such as metamorphic relations, partial oracles, or human judgment—are employed, but correctness is still assumed to be assessable in some form [4, 9]. The effectiveness of automated testing, regression testing, and test adequacy assessment depends on the availability of such oracles to interpret test outcomes meaningfully.

> **A3 — Existence of a Reliable Test Oracle.** For a given test case, there exists a mechanism to determine whether the observed system behavior is correct.

**Assumption A4: Controlled and Stable Execution Environment.** Classical software testing assumes that tests are executed within a *controlled and stable environment* [17]. Foundational testing literature defines a test case not only by its inputs but also by its execution conditions, implicitly assuming that these conditions can be reproduced across test runs [25]. Testing frameworks rely on test harnesses, stubs, and drivers to isolate the system under test from environmental variability and external dependencies [6].

This assumption is critical for regression testing and automated testing pipelines. Regression testing techniques attribute changes in test outcomes to modifications in the system under test, presupposing that the surrounding execution environment remains unchanged [28]. Similarly, continuous integration practices depend on stable build and test environments so that test failures can be interpreted as meaningful signals of regressions rather than artifacts of environmental drift [13].

> **A4 — Controlled and Stable Execution Environment.** Software tests are executed in an environment whose configuration and external dependencies are known, controlled, and stable across test runs.

**Assumption A5: Clear Attribution of Failures.** Classical software testing and debugging assume that observed failures can be *attributed to specific system components or program elements*. Foundational work on debugging treats failure analysis as a process of isolating the code, module, or interaction responsible for an observed incorrect behavior [45]. Testing and debugging techniques are therefore designed to narrow down the set of potential fault locations based on execution traces, coverage information, or component boundaries.

This assumption is embedded in fault localization research and testing practice. Spectrum-based fault localization techniques explicitly rely on the ability to associate test failures with program

elements that are more likely to be faulty [37]. Similarly, object-oriented testing approaches assume that failures can be traced back to individual classes, methods, or interactions, enabling targeted debugging and repair [6].

> **A5 — Clear Attribution of Failures.** When a test fails, the cause of the failure can be localized to specific components, program elements, or interactions within the system.

**Assumption A6: Explicit and Stable Specifications of Correctness.** Classical software testing assumes that the expected behavior of a system is defined by *explicit and stable specifications*. These specifications—derived from requirements, design documents, or contracts—provide the reference against which test outcomes are judged. Foundational testing literature treats correctness as behavior that conforms to stated expectations, enabling test cases to be constructed with well-defined expected results [6, 25].

This assumption underlies many testing activities, including unit testing, system testing, and acceptance testing. More formal approaches, such as design by contract, make this assumption explicit by defining correctness in terms of preconditions, postconditions, and invariants that remain stable across executions and versions [24].

> **A6 — Explicit and Stable Specifications of Correctness.** Correctness is defined by explicit specifications that remain stable enough to support test design and evaluation.

In the remainder of RQ2, we examine how these assumptions hold, weaken, or break down across existing AIware testing approaches.

## 4.2 Mapping Method

To analyze how classical software testing assumptions behave in AIware systems, we systematically map the six baseline assumptions (A1–A6) to the studies included in the retained AIware testing corpus.

For each study, we evaluate every assumption along two orthogonal dimensions: *applicability* and *status*. Applicability captures whether a given assumption is relevant to the system type and testing context considered in the study. When an assumption is not meaningful for a paper's scope or evaluation setting, it is explicitly marked as not applicable rather than being forced into the analysis. For assumptions that are applicable, *status* characterizes how the assumption manifests in the study, including whether it is *upheld*, *implicitly assumed*, *violated*, *adapted*, or *partially holds*. Table 5 summarizes the definitions of the applicability and status labels used in our mapping.

The mapping is performed through close reading of each paper's problem formulation, testing methodology, evaluation design, and stated limitations. For each assumption, we first determine its applicability to the paper's system type and testing context. For assumptions that are applicable or partially applicable, we then assign a status based on how the assumption is treated in the study. An assumption is coded as *implicitly assumed* when it is relied upon without being discussed, and as *violated* or *adapted* when the paper explicitly shows that the classical assumption does not hold or introduces an alternative formulation. When the evidence is ambiguous, we apply a conservative coding strategy, favoring

weaker interpretations (e.g., *implicitly assumed* or *partially holds*) over stronger claims of violation. As the analysis progresses, later studies largely reflect previously identified patterns of assumption breakdown rather than introducing new categories.

## 4.3 Results

Table 6 summarizes how the six classical software testing assumptions behave across the retained AIware testing corpus. Rather than failing uniformly, assumptions break down in different ways: some are repeatedly violated, others hold only under constrained conditions, and several are adapted to accommodate the properties of AIware systems.

*A1 — Deterministic Execution.* Within the retained corpus, this assumption is violated or explicitly adapted in 10 of 16 studies. Multiple studies report substantial run-to-run variability for identical inputs, even under fixed prompts and configurations. Atil et al. [3] show up to 15% accuracy variation and up to 70% best–worst gaps under nominally deterministic LLM settings. Gupta [15] introduces pass@$k$ and demonstrates that pass@1 overestimates agent reliability by 20–40%. As a result, approaches such as repeated sampling and probabilistic consistency metrics replace single-execution evaluation [15, 44]. Test outcomes for AIware systems cannot be treated as stable pass/fail signals, but must be interpreted as distributions over possible behaviors.

*A2 — Stable and Explicit Interfaces.* This assumption is violated or holds only partially in 9 of 16 studies. The effective testing interface in AIware systems includes prompts, contextual information, and tool schemas, all of which may change with model updates or deployment context [33, 40, 44]. Jiang et al. [18] find that API misuse accounts for 32–48% of bugs in LLM libraries, reflecting a shift toward interface-centric defects driven by rapidly evolving APIs. Studies of agentic systems further show that interface behavior drifts due to schema evolution and partial failures [15]. Consequently, tests are bound to volatile interaction surfaces rather than fixed contracts.

*A3 — Existence of a Reliable Test Oracle.* This assumption is violated or adapted in 12 of 16 studies. Binary oracles based on exact output matching are often infeasible due to output variability and open-ended tasks [33, 40]. In response, the corpus exhibits a range of alternative oracle strategies: probabilistic correctness thresholds [44], robustness-based proxies [40], human judgment [20], state-based goal verification for agentic systems [15], and LLM-based judges [30, 42]. For instance, Sorokin et al. [30] operationalize the oracle as a threshold function over LLM-judge scores, defining failure as $(f_1 < 0.75) \vee (f_2 < 0.75)$. These adaptations indicate that correctness assessment in AIware settings is approximate and context-dependent rather than definitive.

*A4 — Controlled and Stable Execution Environment.* This assumption is violated or only partially satisfied in 10 of 16 studies. Execution depends on evolving foundation models, external APIs, and shared infrastructure that may change independently of application code [33, 40]. Ma et al. [23] demonstrate that silent LLM API updates cause prompt regressions without any code change, undermining the classical assumption that test outcome changes reflect system

**Table 5: Definitions of Applicability and Status Labels Used in Assumption Mapping**

| Label | Definition |
|---|---|
| **Applicability** | |
| Applicable | The assumption is relevant to the system type or testing context studied in the paper and can meaningfully be evaluated. |
| Partially Applicable | The assumption applies only under restricted conditions (e.g., benchmarked settings or specific components) or applies to part of the system but not end-to-end. |
| Not Applicable | The assumption is not meaningful for the paper's scope or testing objective and is therefore excluded from evaluation. |
| **Status (for Applicable or Partially Applicable Assumptions)** | |
| Upheld | The assumption holds as in classical software testing and is relied upon without qualification. |
| Implicitly Assumed | The assumption is not discussed explicitly but is required for the proposed testing or evaluation approach to function as intended. |
| Violated | The paper explicitly shows that the assumption does not hold in the studied AIware setting. |
| Adapted | The classical assumption does not hold, and the paper introduces a modified or alternative formulation to replace it. |
| Partially Holds | The assumption holds only under specific constraints (e.g., fixed benchmarks or controlled environments) and does not generalize to broader AIware settings. |
| N/A | Used only when the assumption is marked as Not Applicable. |

modifications. Gupta [15] explicitly injects environmental faults (rate limits, timeouts, schema errors) using chaos-engineering-style fault profiles to model production instability. These findings show that environmental instability is a recurring and structurally distinct source of variability in AIware systems.

*A5 — Clear Attribution of Failures.* This assumption is violated or only partially holds in 11 of 16 studies. Failures often emerge from interactions among prompts, model reasoning, tools, and context rather than isolated code defects [15, 20, 33]. Even when attribution is attempted, it is typically coarse-grained. Winston & Just [35] map failures in tool-augmented LLMs to six root causes spanning the LLM, tool selection, tool set, tool errors, and parsing, demonstrating that repair requires more than simply retraining the model. Most studies identify failure patterns or fault categories rather than precise fault locations [11, 33, 40]. Traditional fault localization techniques require reformulation in AIware settings.

*A6 — Explicit and Stable Specifications of Correctness.* This assumption is violated or adapted in 12 of 16 studies. Correctness is frequently defined using benchmark-specific labels, task-dependent criteria, or proxy measures rather than fixed functional specifications [33, 40, 44]. Tian et al. [32] identify specification and intent defects as a top-level category in their prompt defect taxonomy, showing that ambiguous or underspecified instructions are a primary source of prompt failure. Agent-oriented approaches relax specifications further by using goal states, invariants, or metamorphic relations [15, 20]. Stable and complete specifications are rarely available for AIware systems, requiring more flexible notions of correctness.

Overall, the results in Table 6 show that none of the six classical testing assumptions consistently holds across the retained AIware testing corpus. Instead, AIware systems introduce systematic sources of nondeterminism, interface instability, oracle ambiguity, environmental drift, attribution challenges, and specification uncertainty. These breakdowns motivate our subsequent analysis of lifecycle testing support in RQ3.

**RQ2 Summary**

None of the six classical testing assumptions consistently holds in the AIware testing corpus. Deterministic execution, oracle reliability, and specification stability are the most frequently violated, each affected in 10–12 of 16 studies. Existing work adapts through probabilistic metrics, alternative oracles, and relaxed specifications, but these adaptations remain fragmented.

## 5 RQ3 Analysis and Results

To address **RQ3**, we examine whether existing AIware testing approaches provide adequate support for core testing activities across the software lifecycle. Rather than evaluating individual techniques in isolation, we synthesize the literature by testing level and assess how well the reported approaches support repeatable execution, interpretation of results, and debugging in practice.

### 5.1 Synthesis Approach

This analysis builds directly on the coding dimensions defined in Table 3, namely *Testing Level*, *Test Oracle*, and *Debugging Support*. For each testing level, we aggregate the coded oracle strategies

**Table 6: Breakdown of Classical Software Testing Assumptions in the AIware Testing Corpus**

| Assumption | Role in Classical Testing | Observed Status in AIware | Representative Evidence | Implications for Testing |
|---|---|---|---|---|
| A1 — Deterministic Execution | Enables repeatable execution and regression testing | Violated or adapted in most studies | LLM outputs vary across executions, leading to repeated runs and probabilistic metrics [15, 33, 44] | Test outcomes must be interpreted statistically rather than as binary pass/fail results |
| A2 — Stable and Explicit Interfaces | Defines stable interaction boundaries for test design | Violated or only partially holds | Prompt, context, and tool interfaces change with model updates and usage context [15, 33, 40, 44] | Tests bind to volatile interfaces and must account for drift and interaction variability |
| A3 — Existence of a Reliable Test Oracle | Determines correctness of test executions | Violated or adapted across studies | Binary oracles are replaced by probabilistic, robustness-based, human, or state-based checks [15, 20, 44] | Correctness assessment shifts from exact output matching to approximate or contextual evaluation |
| A4 — Controlled and Stable Execution Environment | Allows failures to be attributed to system changes | Violated or only partially holds | Execution depends on evolving models, APIs, and infrastructure beyond developer control [15, 33, 40, 44] | Regression testing must track environment and model changes, not only code revisions |
| A5 — Clear Attribution of Failures | Supports fault localization and debugging | Violated or only partially holds | Failures emerge from interactions among prompts, models, tools, and context rather than isolated components [15, 20, 33, 40] | Debugging shifts from code-level localization to system-level diagnosis |
| A6 — Explicit and Stable Specifications of Correctness | Provides a fixed reference for expected behavior | Violated or adapted in most studies | Correctness is benchmark-dependent, task-specific, or defined via goal states and invariants [15, 20, 33, 40, 44] | Specifications must be relaxed, contextualized, or reformulated to support testing |

and debugging characteristics and summarize whether the reported workflows allow tests to be executed repeatedly with limited human involvement. The resulting synthesis is shown in Table 7.

Table 7 reorganizes the retained studies by testing level and summarizes the dominant oracle strategies, observed automation characteristics, and diagnostic support at each level. The values shown in the table are derived directly from the RQ1 coding results; no new dimensions are introduced. Representative citations are included to illustrate how these patterns appear in specific studies.

The columns in Table 7 are interpreted as follows. *Dominant oracle strategy* summarizes how correctness or adequacy is assessed (e.g., heuristic rules, human judgment, or goal-based criteria). *Automation readiness (observed)* indicates whether the reported testing workflow can be executed repeatedly with limited human involvement, based on the described execution and oracle mechanisms. *Diagnostic support* reflects whether the approach provides information that helps developers understand or localize failures, as captured in the debugging-support coding.

## 5.2 Results

**System-level testing exposes failures but offers limited lifecycle support.** Within the retained corpus, 13 of 16 studies evaluate AIware applications at the system level through end-to-end assessments of robustness, reliability, or safety. These approaches can reveal high-level failures, but they rely on a mix of exact oracles (7 studies), heuristic oracles (4 studies), metamorphic oracles (1 study), and human judgment (1 study) [15, 20, 38–41]. Although execution is often automated, result interpretation frequently depends on manual judgment or heuristic thresholds. As a result, support for repeatable, automation-ready lifecycle workflows remains uneven.

**Unit-level testing shows more structured execution but limited scope.** One retained study targets unit-level artifacts, focusing on model-level adequacy criteria [42]. This approach supports scripted and repeatable execution, leading to high observed automation readiness. However, its diagnostic capabilities remain limited, and its applicability is restricted to internal model components rather than full application behavior.

**Integration testing is absent in the retained corpus.** Although several studies acknowledge that AIware systems consist of

multiple interacting components [11, 18, 35], none treats integration testing as a first-class activity with instantiated test processes or dedicated oracles. Integration-related challenges are discussed conceptually in taxonomies, without corresponding operational techniques.

**Regression and deployment-time testing remain underdeveloped.** The need to detect behavioral regressions after model or API changes is recognized, but concrete regression testing practices are rare. Only Ma et al. [23] analyze prompt behavior under evolving LLM APIs and frame regression as an important concern, but their work takes the form of analytical and exploratory evaluation rather than stable regression test suites with automated decision criteria. Similarly, while two studies discuss production failures, non-determinism, and reliability issues in deployed systems [3, 33], they do not present CI-integrated or in-production testing workflows with automated oracles and actionable diagnostic outputs.

Overall, the retained corpus emphasizes pre-release, system-level evaluation, with limited support at the unit level. Integration, regression, and deployment-time testing remain sparse or conceptual. Lifecycle-complete testing support comparable to established software engineering practice has not yet emerged.

> **RQ3 Summary**
>
> Existing AIware testing support is concentrated at the system level (13 studies), with mixed automation readiness and limited diagnostic support. Integration, regression, CI-integrated, and in-production testing remain absent or underdeveloped, leaving significant lifecycle gaps.

## 6 Discussion

### 6.1 From Model Evaluation to AIware Testing

Much of the work labeled as testing for AIware systems resembles model evaluation [15, 38–41]. These studies assess robustness, stress behavior, or performance characteristics, but are not designed to support integration, regression, or release validation in a software lifecycle context. While these approaches are valuable, they do not fully align with the traditional role of testing in software engineering.

This helps explain the dominance of system-level testing. The absence of precise specifications and the non-deterministic behavior of AIware components make finer-grained testing difficult. As a result, researchers favor end-to-end evaluation with heuristic or human-based judgments. However, this limits support for regression control, fault localization, and automated quality enforcement.

The closest related work to this survey is Dobslaw et al. [11], which proposes a faceted taxonomy of testing challenges for LLM-based software, organizing challenges around variability sources, oracle ambiguity, and configuration sensitivity. Their contribution is primarily taxonomic and tool-oriented: it characterizes the problem space for testing LLM-based systems. In contrast, the present survey (a) maps existing testing research to classical SE testing levels and lifecycle stages, (b) systematically analyzes how six foundational testing assumptions break down, and (c) synthesizes lifecycle coverage gaps as structural findings. While Dobslaw et al. identify

what makes testing difficult, this survey assesses how existing approaches address—or fail to address—those difficulties through the lens of software engineering practice.

### 6.2 Lifecycle Mismatch in Current AIware Testing Research

RQ3 reveals a clear mismatch between the testing activities emphasized in the AIware literature and those required across the software lifecycle. While classical software engineering treats testing as continuous—spanning development, integration, regression, and deployment—the retained corpus focuses primarily on pre-release, system-level evaluation.

The gap is most visible at the integration level. Although AIware systems involve interacting components, i.e., prompts, orchestration logic, retrieval modules, tools, and model APIs, none of the 16 retained studies operationalizes integration testing with defined workflows and executable oracles. Integration challenges are discussed conceptually, but not instantiated as testing processes [11, 18, 35]. In contrast, classical integration testing frameworks emphasize structured interaction-level validation between components [19, 22].

Regression testing shows a similar limitation. Behavioral drift under model or API updates is widely acknowledged, yet only one study [23] directly addresses regression, and it does so through exploratory comparisons rather than executable regression suites integrated into CI pipelines. This contrasts with established regression practices in software engineering [12, 29, 43].

Deployment-time testing is also weakly supported. While two studies analyze production-like failures and non-determinism [3, 33], they do not present CI-integrated or in-production testing workflows with automated decision criteria. Even stress-testing work remains experimental rather than lifecycle-integrated [15].

Overall, AIware testing research emphasizes exposing system-level failures but provides limited support for long-term maintenance and evolution.

### 6.3 Oracle Design as the Central Bottleneck

Across testing levels and lifecycle stages, the dominant limitation is the absence of reliable and automatable test oracles. While many approaches can generate inputs and execute tests at scale, determining whether observed behavior constitutes a failure remains difficult. This aligns with RQ2, which showed that deterministic execution and exact correctness rarely hold in AIware systems.

Unlike traditional software testing, where pass/fail conditions are derived from specifications [14], AIware systems exhibit non-determinism, output variability, and underspecified correctness. Most approaches therefore rely on heuristic, probabilistic, or human-in-the-loop oracles [4, 10, 34]. Empirical studies further show that identical executions can yield divergent outputs [3].

A notable emerging strategy is the use of LLM-based judges as automated oracles. Within the retained corpus, Sorokin et al. [30] employ an LLM-based judge to evaluate response quality, using continuous scoring during search and binary classification post-search. Xie et al. [42] use GPT-judge and LLaMA-Guard to detect hallucination and toxicity. Within the oracle taxonomy defined in Table 3,

**Table 7: Lifecycle-oriented synthesis of AIware testing support derived from Table 3 coding. Representative studies are cited per testing level to illustrate dominant oracle strategies, automation characteristics, and diagnostic support observed in the corpus.**

| Testing Level | Dominant Oracle Strategy | Automation Readiness (Observed) | Diagnostic Support |
|---|---|---|---|
| Unit | Heuristic [42] | High (scripted, offline execution) | Limited |
| Integration | Absent (no primary studies) [11] | None | None |
| System | Heuristic; Human; Exact (goal-based) [15, 20, 38, 40] | Mixed (automated execution with manual or heuristic oracles) | Limited-Strong |
| Regression | Exact [23] | Low (unstable, ad hoc analyses) | None |
| CI / In-production | Not established [15, 33] | None | None |

LLM-as-a-Judge functions as a heuristic oracle: it provides a scalable and automated correctness signal, but one that is approximate and subject to its own limitations. In particular, LLM-based judges introduce second-order oracle dependencies—the judge model itself may be non-deterministic, sensitive to prompt framing, or subject to version drift, compounding the oracle challenges already present in AIware testing. Despite these limitations, LLM-as-a-Judge represents a practical path toward automatable oracles and may be particularly relevant for closing the CI and deployment-time testing gaps identified in RQ3, where scalable automated evaluation is a prerequisite.

This oracle ambiguity limits automation more broadly. Without stable pass/fail criteria, CI-based gating, automated regression checks, and systematic failure triage become unreliable. The lack of CI-integrated testing in the corpus reflects this constraint rather than just a tooling gap.

Existing approaches, such as adequacy metrics, diversity criteria, and metamorphic relations [9, 42, 44], provide approximate signals but limited diagnostic insight. Regression studies similarly rely on aggregate comparisons rather than executable test suites with clear failure conditions [23].

Advancing AIware testing beyond pre-release evaluation will require new oracle designs that are both executable and tolerant to non-determinism.

### 6.4 Implications for Researchers

The results of RQ2 and RQ3 suggest that future progress depends less on additional system-level testing techniques and more on addressing lifecycle coverage, oracle design, and diagnostic support.

*Integration testing as a first-class problem.* Future work should explicitly target interactions among prompts, orchestration logic, retrieval modules, tools, and external APIs. Making integration testing operational would enable earlier detection of interaction-level failures.

*Regression testing under non-determinism.* Regression should move beyond exploratory comparison toward executable strategies that tolerate non-determinism while still providing meaningful signals [23].

*Oracle design beyond detection.* Future oracle designs should support both failure detection and diagnosis. Existing surrogate mechanisms—adequacy metrics, diversity criteria, metamorphic relations, and LLM-based judges—offer detection but limited explanatory power [9, 30, 42, 44].

*Lifecycle-aware workflows.* AIware testing needs to evolve from isolated experimental evaluation to continuous, lifecycle-integrated workflows that support versioning, deployment, and evolution.

### 6.5 Implications for Practitioners

Current AIware testing techniques are most effective at pre-release, system-level evaluation. Practitioners can adopt robustness testing, stress testing, and human-in-the-loop evaluation to expose high-risk behaviors before deployment.

However, traditional CI-style automated pass/fail gating is difficult to implement due to unstable oracles and non-determinism. Automated regression triggered by code changes remains unreliable in many AIware settings.

Practitioners should therefore treat AIware testing as part of a broader risk management strategy, combining pre-release testing with monitoring, logging, and post-deployment analysis. Since reliable pass/fail guarantees are often not feasible, teams should also rely on runtime monitoring, safeguards, and human review to manage risk.

### 7 Threats to Validity

As with any survey-based study, our findings are subject to several validity threats. We summarize these threats and the steps taken to mitigate them.

*Study selection and search scope.* A primary threat concerns the completeness of the corpus. Our search focused on established software engineering and AI venues to capture methodologically mature work [21, 36]. This may exclude relevant studies published in lower-ranked or emerging venues. To mitigate this risk, we complemented database searches with backward and forward snowballing and conducted a final gap-oriented verification pass. Nevertheless, some niche or very recent studies may not have been captured.

*Coding subjectivity and interpretation bias.* The analysis relies on manual coding of testing level, oracle type, and debugging support. As described in Section 2.2, coding was performed by one researcher with a second researcher independently reviewing all decisions; disagreements were resolved collaboratively. Single-annotator coding nonetheless introduces the risk of subjective interpretation, particularly when studies use informal terminology [36]. To reduce bias, we applied a conservative strategy, assigning a single primary value

per dimension and coding only what was explicitly described. Although this process reduces bias, alternative interpretations remain possible in borderline cases.

*Interpreting absence as a research gap.* Our identification of gaps —particularly in integration, regression, and CI-integrated testing—relies on the absence of concrete testing methods in the retained corpus. It is possible that relevant methods exist under different terminology or are embedded within broader frameworks. To mitigate this risk, we cross-referenced our findings with recent taxonomies and surveys on AI/ML system failures and testing [11, 16], which report similar lifecycle coverage limitations.

*Rapid evolution of the field.* AIware testing is evolving quickly, and many studies appear first as preprints. Findings may change as work matures. To reduce this temporal threat, we focus on methodological characteristics and testing concepts rather than reported performance outcomes, which are more likely to shift.

*Generalizability of conclusions.* Our conclusions describe trends observed in the retained corpus (N=16) rather than evaluating the effectiveness of individual techniques. The relatively small size of the corpus reflects the emerging nature of AIware testing research. Findings should therefore be interpreted as indicative of current research patterns within this scoped set of studies, rather than as exhaustive coverage of all possible approaches.

Overall, while these threats cannot be eliminated, the systematic search strategy, conservative coding approach, and cross-validation with prior surveys support the credibility of our conclusions.

## 8 Conclusion

This survey examines the current state of testing research for AIware systems, with a focus on how existing approaches align with established software testing concepts and software lifecycle practices. By analyzing the literature across testing levels, oracle types, and lifecycle stages, we identify clear patterns in both the emphasis and the limitations of current AIware testing research.

The results show that most existing work focuses on system-level testing and pre-release evaluation. While this focus reflects the challenges introduced by non-deterministic and underspecified behavior, it also leaves important gaps in areas that are central to software engineering practice, such as integration testing, regression testing, and CI-based quality control. These gaps are not incidental; they arise from fundamental difficulties in defining reliable test oracles and supporting diagnosis in AIware systems.

Our lifecycle-oriented analysis highlights oracle design as a central constraint on the practical use of AIware testing techniques. Without oracles that support automation and debugging, many approaches remain limited to exploratory evaluation and cannot be reliably integrated into development and maintenance workflows. As a result, current testing research often falls short of supporting long-term system evolution.

Overall, this survey clarifies where AIware testing research currently stands and outlines the structural challenges that must be addressed to support lifecycle-aware, reliable AIware systems.

# A  Appendix

## A.1  RQ1 Coding Examples

This section provides illustrative examples of the per-paper coding process used in our analysis. For transparency, we present detailed extraction tables for two representative papers from the AIware corpus. These examples demonstrate how each coding dimension was derived based on explicit evidence from the respective paper.

**Table A1: Example of the per-paper coding process for Dobslaw et al. [11], illustrating how coding dimensions were assigned and grounded in explicit evidence from the study.**

| Coding Dimension | Assigned Value | Description | Evidence Location in the Paper |
|---|---|---|---|
| Testing level | System (conceptual) | The paper discusses testing challenges at the level of complete LLM-based software executions, focusing on variability, non-determinism, and test outcomes across runs. It does not describe unit- or component-level tests, but instead frames testing at the level of full system behavior. | 1. Introduction; 3.3 Goal; 5. Discussion |
| AIware type | Agentic | The study targets software systems that embed LLMs, including single-LLM and multi-agent LLM systems. The focus is on application-level behavior rather than standalone model evaluation or testing tools. | 1. Introduction; 2. Background and Motivation |
| Testing Technique | Conceptual / Taxonomy-based analysis | Rather than executing tests, the paper organizes testing challenges into facets describing variability sources, oracle ambiguity, and test design concerns. No executable testing framework or algorithm is proposed. | 3.2 Software Under Test; 3.5 Inputs; 4.1 Manual Taxonomy-Based Tool Evaluation |
| Test oracle | Heuristic | Correctness is framed as aggregated judgment across multiple executions due to inherent non-determinism. The taxonomy distinguishes atomic versus aggregated oracle perspectives and explicitly rejects single expected outputs as sufficient. | 3.4 Oracles; 4.3.1 Non-deterministic Outputs and Usage of Aggregated Oracle |
| Debugging support | Limited | The paper identifies sources of testing difficulty and variability but does not provide debugging workflows, fault localization, or repair strategies. Its contribution is explanatory rather than operational. | 4.1 Manual Taxonomy-Based Tool Evaluation; 4.3 LLM sensitivity analysis; 5. Discussion |
| Failure Types | Variability / Oracle Ambiguity | Failures are not defined as crashes or incorrect outputs, but as situations where non-determinism, variability, or configuration sensitivity make test outcomes ambiguous or inconclusive. | Abstract; 4.2 LLM-based Tool Evaluation Through a Detailed Checklist/Prompt Taxonomy Facets |

**Table A2: Example of the per-paper coding process for Ma et al. [23], illustrating how coding dimensions were assigned and grounded in explicit evidence from the study.**

| Coding Dimension | Assigned Value | Description | Evidence Location in the Paper |
|---|---|---|---|
| Testing Level | System-level regression analysis (slice-based) | Compares model behavior before/after API updates; defines regression over slice-level aggregated metrics rather than single predictions | Abstract; 1 Introduction; 3.2 Observations; 4.1 Identifying Data Slices as Regression Test Suites |
| AIware Type | Prompt-based system | LLM API + prompt templates; no tools or multi-agent workflow | 2.2 The Rise of Prompting LLMs; 3 Case Study: Toxicity Detection |
| Testing Technique | Exploratory slice-based regression analysis | Cross-version accuracy/F1 comparison; slice-level performance tracking; entropy-based uncertainty analysis | 3.1 Experiment Setup; 3.2 Observations; 4 Towards Regression Testing for Prompting LLMs |
| Test Oracle | Metric-based (heuristic) | Accuracy and F1 used; regression defined over slice-level aggregated metrics rather than individual prediction flips | 3.1.4 Metrics; 4.1 Identifying Data Slices as Regression Test Suites |
| Debugging Support | Limited | Identifies regressions and affected slices; recommends tracking prompts but no fault localization or repair workflow | 3.2 Observations; 4.2 Tracking Prompts for Regression Testing |
| Execution Context | Offline experimental study | Fixed datasets; repeated API calls; no CI/CD or deployment-time monitoring | 3.1 Experiment Setup |
| Automation Support | Partial | Automated metric computation and entropy estimation; no automated regression threshold framework or CI pipeline | 3.1 Experiment Setup; 3.2 Observations |
| System Scope | End-to-end prompt + API behavior | Evaluates final classification outputs only; no internal model component inspection | 3 Case Study: Toxicity Detection |

## A.2    RQ2 Coding Examples

To illustrate how classical testing assumptions were mapped in our analysis, we provide two representative assumption-level coding examples for two papers. These tables show how each assumption (A1–A6) was evaluated in context, including its applicability, observed treatment, and supporting evidence.

### Table A3: RQ2: Assumption-level coding example for Xiao et al. [40]

| Assumption | Status in P002 | Applicability | What P002 Observes / Does | Why the Assumption Is Problematic in AIware | Concrete Evidence from P002 |
|---|---|---|---|---|---|
| A1 — Deterministic Execution | Implicitly Assumed | Applicable | Treats robustness failures as reproducible under fixed perturbations | LLM outputs remain stochastic, but variability is not explicitly addressed | No repeated executions per test case; success rate computed per generated input |
| A2 — Stable & Explicit Interfaces | Upheld | Applicable | Treats prompt+example pair as a unified, stable input interface | Interface abstraction holds only as long as model behavior is stable | Defines testing unit as "input prompts and examples as a unified whole" |
| A3 — Reliable Test Oracle | Upheld | Applicable | Uses label-preserving robustness oracle (prediction change under perturbation) | Correctness conflated with robustness; semantic correctness not reassessed | Success defined via prediction inconsistency under synonym substitution |
| A4 — Controlled Execution Environment | Partially Holds | Partially Applicable | Fixes model versions (LLaMA2-13B / 70B) and datasets | Environment control depends on frozen models and offline execution | Explicitly evaluates transferability across model sizes |
| A5 — Clear Attribution of Failures | Partially Holds | Partially Applicable | Attributes failures to specific word-level perturbations | Attribution remains heuristic, not causal | Adaptive WIR identifies "important" words driving failure |
| A6 — Explicit & Stable Specifications | Upheld | Applicable | Uses labeled NLP datasets with ground-truth labels | Specification stability depends on task framing and dataset quality | Evaluates on MR, AG's News, etc. with fixed labels |

### Table A4: RQ2: Assumption-level coding example for Gupta et al. [15].

| Assumption | Status of Assumption in P005 | Applicability | What P005 Observes / Does | Why the Assumption Is Problematic in AIware | Concrete Evidence from P005 |
|---|---|---|---|---|---|
| A1 — Deterministic Execution | Violated | Applicable | Treats execution as inherently stochastic; evaluates consistency via passk | Single-run outcomes systematically overestimate reliability | Introduces passk and shows pass@1 overestimates reliability by 20–40% |
| A2 — Stable & Explicit Interfaces | Violated | Applicable | Treats prompts, tools, and schemas as volatile interaction surfaces | Interfaces evolve via schema drift, paraphrasing, and tool failures | Fault profiles include schema drift, partial responses, API changes |
| A3 — Reliable Test Oracle | Adapted | Applicable | Replaces output-based oracles with state-based goal verification | Textual correctness is insufficient for agentic tasks | Defines correctness via deterministic state predicates (Alg. 1) |
| A4 — Controlled Execution Environment | Violated | Applicable | Explicitly injects faults to model production instability | Real deployments include network, rate-limit, and infra failures | Chaos-engineering-style fault injection with $\lambda$-profiles |
| A5 — Clear Attribution of Failures | Partially Holds | Partially Applicable | Attributes failures to fault types rather than code locations | Agent failures emerge across reasoning, tools, and recovery logic | Fault ablation isolates rate-limit vs timeout vs schema-drift impacts |
| A6 — Explicit & Stable Specifications | Adapted | Applicable | Defines correctness as goal-state satisfaction under perturbation | Specs must tolerate paraphrase, reordering, and correction | Action Metamorphic Relations preserve end-state equivalence |

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
