# OpenReview forum: "Testing AIware Systems: A Software Engineering Survey"
_ACM.org/AIWare/2026/Conference — AIware 2026_

### Official Review · Reviewer_g59b · 2026-02-24

**Rating:** 3
**Confidence:** 5

**Review:**

(+) Strength:

+ The survey is structured around classical software engineering concepts, which provide a clear and useful analytical lens.

+  The authors provide a replication package, which supports transparency and reproducibility.

(-) Weaknesses:

- The survey methodology lacks sufficient detail, particularly regarding search strategy, study selection, and screening procedures.

- The justification for focusing only on system-level work is not fully explained.

(I) Detailed Comment:

- The introduction does not clearly explain the motivation behind conducting the study. I would recommend strengthening the problem statement and explicitly identifying the research gap to improve clarity.

- Include refs: “In response, a growing body of work proposes testing techniques for AIware systems, including robustness testing, search-based test generation, metamorphic testing [9], and human-in-the-loop evaluation.”

- Provide further explanation of why the survey scope is just system level: “The scope of this survey is intentionally narrow. We focus on work that addresses testing or evaluation from a system-level software engineering perspective.”

- Include the full list of selected studies in the main text.

- The survey methodology needs more detail. Clarify the following points: (i) the exact search queries used; (ii) why only the top 10 search results were screened; (iii) who participated in the study selection process; and (iv) the number of papers included and excluded at each phase of screening.

- The authors provided a replication package that supports transparency and reproducibility.

- Consider adding a summary box synthesizing the main findings of each RQ to improve readability.

- Include refs: “Much of the work labeled as testing for AIware systems resembles model evaluation.” And “ Many studies assess robustness, stress behavior, or performance characteristics...”

- Minor:

- Use acronyms consistently (e.g., FMs and LLMs) after defining them once.

-  Page 3, line 278: fix the incorrect figure reference (“Figure ??”).

**Summary:**

The study presents a survey that analyzes the AIware testing literature, examining testing levels, oracle strategies, automation readiness, and diagnostic support, to assess how existing approaches map to lifecycle activities such as integration testing, regression testing, and CI-integrated workflows.

---

> ### Author Response · Authors · 2026-03-20
>
> We thank the reviewer for the detailed and constructive feedback. We have revised the paper to address each concern. Below we describe the specific changes made.
>
> **1. Strengthening the Introduction and Problem Statement**
>
> We have revised the introduction to make the research gap more explicit: prior surveys address model-level testing [46] or behavioral NLP testing [27], but neither examines how testing research maps to software engineering lifecycle practices when FMs are embedded within software systems. We have also added the missing references in the testing techniques passage (now citing corpus studies for each technique category) and in the model evaluation discussion in Section 6.1.
>
> **2. Clarification of Survey Scope**
>
> We appreciate this comment, as it identified an ambiguity in our phrasing. Our survey scope is *not* restricted to system-level testing. The phrase "system-level software engineering perspective" was intended to distinguish our focus—testing of AIware as software systems—from model-only evaluation. The system-level concentration is an empirical *finding*, not a scope restriction. The RQ3 analysis (Table 6) explicitly covers unit, integration, regression, and CI/in-production levels and documents their absence or under-representation.
>
> We have revised the scope statement in Section 2.1 to eliminate this ambiguity. The revised text now reads: *"The scope of this survey covers testing and evaluation of AIware systems from a software engineering perspective. This includes studies that propose or analyze concrete testing activities at any level (unit, integration, system, regression, or deployment) provided they target software systems that embed FMs, as opposed to model-only evaluation or benchmark-only assessment."*
>
> **3. Survey Methodology Details**
>
> We have added the following to Section 2.2:
>
> *(i) Search queries.* We used 15 keyword queries in Phase 1, organized into three families: testing/debugging-oriented (e.g., "LLM application testing," "debugging LLM systems"), system/architecture-oriented (e.g., "agentic LLM evaluation," "tool-using LLM evaluation"), and reliability/deployment-oriented (e.g., "LLM reliability production," "LLM system robustness"). Phase 3 added three gap-targeted queries (e.g., "Regression testing LLM-based software"). The complete list is in the replication package.
>
> *(ii) Top-10 justification.* The top-10 limit applies only to keyword-based discovery (Phases 1 and 3). We acknowledge that this cutoff may miss relevant work ranked lower in search results. However, the keyword phases serve as seed identification and gap verification, not as the sole discovery mechanism. The majority of the corpus (14 of 21 candidates) was identified through Phase 2 citation expansion, which imposed no result-count limit and screened 479 candidates. Thematic saturation was observed within the top 10 keyword results, and Phase 3 confirmed that no new testing concepts emerged from targeted gap searches.
>
> *(iii) Study selection process.* Coding was performed by one researcher; a second researcher independently reviewed all decisions, with disagreements resolved collaboratively. This description has been moved from Section 7 to Section 2.2.
>
> *(iv) Per-phase screening numbers.* Phase 1: 150 candidates screened, 7 retained. Phase 2: 479 candidates screened, 14 retained. Phase 3: 30 candidates screened, 0 retained—confirming the identified gaps as substantive. Total: 662 candidates screened, 16 primary studies retained.
>
> **4. Full List of Selected Studies**
>
> We have added a corpus summary table (Table 4) listing all 16 studies with citation, year, venue type, primary testing level, and system type.
>
> **5. RQ Summary Boxes**
>
> We have added a concise summary box at the end of each RQ section (Sections 3, 4, and 5).
>
> **6. Missing References**
>
> Added in both indicated passages (Section 1 and Section 6.1).
>
> **7. Minor Issues**
>
> Acronym consistency (FM/LLM) has been fixed throughout. The "Figure ??" reference has been corrected to "Figure 1."
>
> We believe these revisions address all concerns raised and strengthen the paper's methodological transparency and clarity. We thank the reviewer for the thorough engagement.

---

### Official Review · Reviewer_J6Kh · 2026-03-09

**Rating:** 4
**Confidence:** 3

**Review:**

# Strengths

- Targeting an important area for AIWare development
- Well-written paper (clear breakdowns, observations and implications)


# Weaknesses

- Scope of "AIware" system in this paper needs a bit clarification
- No discussion on LLM-as-a-Judge for evaluating AIware systems

# Detailed comments to the authors

This paper is well-written, I appreciate it discusses its study review and scope in detail, and presents a clear breakdown on the study corpus with useful observations and actionable implications.

Some minor questions are:

I want to get a clearer understanding on what is considered as an AIware system in this paper, e.g., how would you consider when an LLM is a core component of the system? If a rather traditional system that uses a classical ML model had its ML model replaced with an LLM and some minimal support for using the LLM, does it count as AIware in this paper?  Do you see if some of these assumptions from classical testing be violated more or less (if those are counted as AIware)?

It seems no work related to LLM-as-a-Judge is discussed. How do studies that use LLM-as-a-Judge and LLM-generated specifications/rubrics for rating agent runtime behaviors and outcomes in offline and online deployment fit or do not fit into this paper? Do you consider those as some form of testing/evaluation or where they should fit into the testing cycle in the AIware era?

Typo: L278, L863

# Questions to the authors

NA

**Summary:**

This paper focuses on surviving testing on systems that use LLM as a core component. It discusses study curation scope and process, characterize the curated study corpus, identify assumptions held in testing of classical software engineering, examine whether these assumptions held for LLM-centric systems, and discuss main observations and implications.

---

> ### Author Response · Authors · 2026-03-20
>
> We thank the reviewer for the positive assessment and the thoughtful questions. We have revised the paper to address both points.
>
> **1. Clarifying the AIware System Boundary**
>
> This is an important question. We have added a discussion in Section 2.1 to clarify the boundary. Under our definition, a system qualifies as AIware when a foundation model is embedded as a core component whose probabilistic and generative properties influence system behavior. A traditional system that replaces a classical ML model with an LLM behind a fixed, deterministic adapter (e.g., returning only a top-1 classification label at temperature 0) falls at the boundary of our scope.
>
> The critical differentiator is the degree to which the FM's probabilistic nature propagates through the system. In a constrained drop-in setting, assumptions such as deterministic execution (A1) and oracle reliability (A3) may be less severely affected, because the model's stochastic behavior is not exposed to the rest of the system. By contrast, systems where the FM participates in open-ended generation, multi-step reasoning, or tool invocation expose more of the properties that drive assumption breakdown. The degree of assumption violation thus varies along a spectrum tied to how much of the FM's behavior is exposed to the surrounding architecture.
>
> **2. LLM-as-a-Judge**
>
> We appreciate this observation. LLM-as-a-Judge is indeed present in our corpus, though it was not discussed as a distinct category in the original submission. Sorokin et al. (STELLAR) use an LLM-based judge to evaluate response quality with continuous scoring during search and binary classification post-search. Xie et al. (LeCov) use GPT-judge and LLaMA-Guard for hallucination and toxicity detection.
>
> We have added a dedicated discussion in Section 6.3 (Oracle Design as the Central Bottleneck). Within our oracle taxonomy, LLM-as-a-Judge functions as a heuristic oracle: it provides a scalable and automated correctness signal, but one that is approximate. Importantly, it introduces second-order oracle dependencies, i.e., the judge model itself may be non-deterministic, sensitive to prompt framing, or subject to version drift, compounding the oracle challenges already present in AIware testing. Despite these limitations, LLM-as-a-Judge represents a practical path toward automatable oracles and may be particularly relevant for closing the CI and deployment-time testing gaps identified in RQ3, where scalable automated evaluation is a prerequisite.
>
> We consider this a valuable addition and thank the reviewer for raising it.
>
> **3. Typos**
>
> Typos have been fixed.
>
> We thank the reviewer again for the thorough assessment of our work.

---

### Official Review · Reviewer_EtC6 · 2026-03-10

**Rating:** 3
**Confidence:** 3

**Review:**

**Pros**
- The topic addressed in the paper is timely and relevant.
- The paper fits well within the scope of the conference.

**Cons**
- Some aspects of the methodology are not sufficiently explained or justified.
- The paper provides limited quantitative support for several observations.
- The discussion includes few concrete examples from the surveyed papers.


**Evaluation**
1. While I understand that screening papers is a time-consuming and labor-intensive process, I do not understand why the authors restrict themselves to only the top ten papers when identifying the initial seed set. This approach could be reasonable if the purpose of the seed papers were simply to extract keywords for the subsequent search. However, in this case, the seed papers are used for backward and forward snowballing. In my view, limiting the seed set in this way may unnecessarily restrict the survey's search space and could lead to the omission of relevant work.
2. Another limitation of the paper is the limited of transparency in the study process. In particular, the following aspects are not adequately explained:
   - It is unclear whether the keywords used in Phase 1 are exclusively those reported in Section 2.2, or whether those are only illustrative examples.
   - Phase 3 is only briefly described. For example, expressions such as "targeted searches using focused queries" are too vague and do not help the reader understand the actual process or the keywords employed.
   - The authors do not provide quantitative details on the filtering steps, such as how many papers were considered and excluded in each phase. I understand that some of this information is available in the replication package, but I would recommend including it directly in the paper.
   - If space permits, I would also encourage the authors to include the full list of papers in the final corpus. If additional space is needed, Section 4.1 could perhaps be made more compact.

3. The authors often do not provide quantitative support for the survey findings. For example, they frequently refer to "several studies" or "some studies" without specifying how many. As a result, the discussion lacks quantitative grounding.
4. Although the paper provides useful insights into the current state of testing for LLM-based systems, it rarely includes concrete examples from the surveyed studies. This often makes the discussion too abstract and prevents the reader from gaining more tangible insights into the corpus and its contents.
5. In my opinion, the paper is novel, as I am not aware of other surveys specifically targeting the testing of AIware systems. The closest related work is probably the paper by Dobslaw et al. [11], which is also part of the survey corpus. It would have been helpful if the authors had discussed this paper more explicitly and highlighted the main differences with respect to the present work.
6. The paper is generally well presented. I only noticed one minor issue: on page 3 line 278 "Figure ??" should be replaced with "Figure 1".

**Summary:**

The paper surveys the existing literature on testing LLM-based software systems (i.e., AIware systems). In particular, the authors examine to what extent assumptions from classical software testing still hold for this new class of systems, characterize the current state of research on AIware testing, and highlight gaps that should be addressed in future work.

---

> ### Author Response · Authors · 2026-03-20
>
> We thank the reviewer for the constructive and detailed feedback. We have revised the paper to address each concern.
>
> **1. Top-10 Seed Set Limitation**
>
> We acknowledge that the top-10 cutoff in keyword-based phases may miss relevant work ranked lower in search results. However, the keyword phases (Phases 1 and 3) serve as seed identification and gap verification, not as the sole discovery mechanism. The majority of the corpus (14 of 21 retained candidates) was identified through Phase 2 citation expansion, which imposed no result-count limit: backward citation search examined full reference lists (averaging ~40 references per seed paper, totaling 479 candidates screened), forward citation search covered all available citing papers, and similarity-based expansion reviewed related-article suggestions. Thematic saturation was observed within the top 10 keyword results, and Phase 3 confirmed that no new testing concepts emerged from targeted gap searches. This justification has been added to Section 2.2.
>
> **2. Methodology Transparency**
>
> We have added the following details directly to Section 2.2:
>
> - *Keywords:* Phase 1 used 15 queries organized into three families: testing/debugging-oriented (e.g., "LLM application testing"), system/architecture-oriented (e.g., "agentic LLM evaluation"), and reliability/deployment-oriented (e.g., "LLM reliability production"). Phase 3 used three gap-targeted queries: "Regression testing LLM-based software," "continuous integration testing LLM application," and "debugging OR fault localization LLM-based system." The complete list is in the replication package.
> - *Phase 3 detail:* The corpus was classified by testing level, underrepresented levels were identified (regression, CI, debugging/fault localization), and three targeted queries were run. Phase 3 screened 30 candidates and retained 0 new papers, confirming the gaps as substantive.
> - *Per-phase filtering:* Phase 1: 150 screened → 7 retained. Phase 2: 479 screened → 14 retained. Phase 3: 30 screened → 0 retained. Total: 662 candidates screened, 16 primary studies retained.
> - *Coding process:* One researcher performed coding; a second researcher independently reviewed all decisions with disagreements resolved collaboratively. This has been moved from Section 7 to Section 2.2.
>
> **3. Quantitative Support**
>
> We have replaced all vague quantifiers with exact counts throughout. For example: "the majority conduct system-level evaluation" → "13 of 16 studies conduct system-level evaluation"; "A substantial portion" → "Four studies"; "Several studies" → specific counts with citations. This applies across Sections 3, 4, and 5.
>
> **4. Concrete Examples from Surveyed Papers**
>
> We have added concrete examples throughout the RQ2 analysis. For instance: for A1 (determinism), Atil et al. show up to 15% accuracy variation under nominally deterministic settings, and Gupta demonstrates that pass@1 overestimates reliability by 20–40%. For A3 (oracles), Sorokin et al. operationalize the oracle as a threshold function over LLM-judge scores. For A5 (attribution), Winston & Just map failures to six root causes spanning LLM, tool selection, and parsing components. Similar concrete evidence has been added for each assumption.
>
> **5. Explicit Comparison with Dobslaw et al. [11]**
>
> We have added a dedicated paragraph in Section 6.1. Dobslaw et al. propose a faceted taxonomy of testing challenges, organizing them around variability sources, oracle ambiguity, and configuration sensitivity. In contrast, our survey (a) maps testing research to classical SE testing levels and lifecycle stages, (b) systematically analyzes how six foundational testing assumptions break down, and (c) synthesizes lifecycle coverage gaps as structural findings. While Dobslaw et al. characterize the problem space, we assess how existing approaches address those difficulties through the lens of software engineering practice.
>
> **6. Full Corpus List**
>
> We have added a corpus summary table (Table 4) listing all 16 retained studies with citation, title, year, venue type, primary testing level, and system type.
>
> **7. Figure Reference**
>
> The "Figure ??" on line 278 has been corrected to "Figure 1."
>
> We thank the reviewer for the thorough engagement with our work.